# Making Efficient Use of Demonstrations to Solve Hard Exploration Problems

**Caglar Gulcehre**\*, **Tom Le Paine**\*, **Bobak Shahriari**, **Misha Denil**, **Matt Hoffman**,
**Hubert Soyer**, **Richard Tanburn**, **Steven Kapturowski**, **Neil Rabinowitz**, **Duncan Williams**,
**Gabriel Barth-Maron**, **Ziyu Wang**, **Nando de Freitas**, **Worlds Team**
DeepMind

## Abstract

This paper introduces R2D3, an agent that makes efficient use of demonstrations to solve hard exploration problems in partially observable environments with highly variable initial conditions. We also introduce a suite of eight tasks that combine these three properties, and show that R2D3 can solve several of the tasks where other state of the art methods (both with and without demonstrations) fail to see even a single successful trajectory after tens of billions of steps of exploration.

## 1 Introduction

Reinforcement learning from demonstrations has proven to be an effective strategy for attacking problems that require sample efficiency and involve hard exploration. For example, Aytar et al. (2018), Pohlen et al. (2018) and Salimans and Chen (2018b) have shown that RL with demonstrations can address the hard exploration problem in Montezuma's Revenge. Večerík et al. (2017), Merel et al. (2017) and Paine et al. (2018) have demonstrated similar results in robotics. Many other works have shown that demonstrations can accelerate learning and address hard-exploration tasks (e.g. see Hester et al., 2018; Kim et al., 2013; Nair et al., 2018; Kang et al., 2018).

In this paper, we attack the problem of learning from demonstrations in hard exploration tasks in partially observable environments with highly variable initial conditions. These three aspects together conspire to make learning challenging:

1. **Sparse rewards** induce a difficult exploration problem, which is a challenge for many state of the art RL methods. An environment has sparse reward when a non-zero reward is only seen after taking a long sequence of correct actions. Our approach is able to solve tasks where standard methods run for billions of steps without seeing a single non-zero reward.

2. **Partial observability** forces the use of memory, and also reduces the generality of information provided by a single demonstration, since trajectories cannot be broken into isolated transitions using the Markov property. An environment has partial observability if the agent can only observe a part of the environment at each timestep.

3. **Highly variable initial conditions** (i.e. changes in the starting configuration of the environment in each episode) are a big challenge for learning from demonstrations, because the demonstrations can not account for all possible configurations. When the initial conditions are fixed it is possible to be extremely efficient through tracking (Aytar et al., 2018; Peng et al., 2018); however, with a large variety of initial conditions the agent is forced to generalize over environment configurations not present in demonstrations. Generalizing between different initial conditions is known to be difficult (Ghosh et al., 2017; Langlois et al., 2019; Zolna et al., 2019).

Our approach to these problems combines demonstrations with off-policy, recurrent Q-learning in a way that allows us to make very efficient use of the available data. In particular, we vastly outperform behavioral cloning using the same set of demonstrations in all of our experiments.

---

\*indicates joint first authorship, both authors equally contributed to this project.

Another desirable property of our approach is that our agents are able to learn to outperform the demonstrators, and in some cases even to discover strategies that the demonstrators were not aware of. In one of our tasks the agent is able to discover and exploit a bug in the environment in spite of all the demonstrators completing the task in the intended way.

Learning from a small number of demonstrations under highly variable initial conditions is not straight-forward. We identify a key parameter of our algorithm, the *demo-ratio*, which controls the proportion of expert demonstrations vs agent experience in each training batch. This hyper-parameter has a dramatic effect on the performance of the algorithm. Surprisingly, we find that the optimal demo ratio is very small (but non-zero) across a wide variety of tasks.

The mechanism our agents use to efficiently extract information from expert demonstrations is to use them in a way that guides (or biases) the agent's own autonomous exploration of the environment. Although this mechanism is not obvious from the algorithm construction, our behavioral analysis confirms the presence of this guided exploration effect.

To demonstrate the effectiveness of our approach we introduce a suite of tasks (which we call the *Hard-Eight* suite) that exhibit our three targeted properties. The tasks are set in a procedurally-generated 3D world, and require complex behavior (e.g. tool use, long-horizon memory) from the agent to succeed. The tasks are designed to be difficult challenges in our targeted setting, and several state of the art methods (themselves ablations of our approach) fail to solve them.

The main contributions of this paper are, firstly we design a new agent that makes efficient use of demonstrations to solve sparse reward tasks in partially observed environments with highly variable initial conditions. Secondly, we provide an analysis of the mechanism our agents use to exploit information from the demonstrations. Lastly, we introduce a suite of eight tasks that support this line of research.

## 2 RECURRENT REPLAY DISTRIBUTED DQN FROM DEMONSTRATIONS

We propose a new agent, which we refer to as Recurrent Replay Distributed DQN from Demonstrations (R2D3). R2D3 is designed to make efficient use of demonstrations to solve sparse reward tasks in partially observed environments with highly variable initial conditions. This section gives an overview of the agent, and detailed pseudocode can be found in Section 2.1.

The architecture of the R2D3 agent is shown in Figure 1. There are several actor processes, each running independent copies of the behavior against an instance of the environment. Each actor streams its experience to a shared

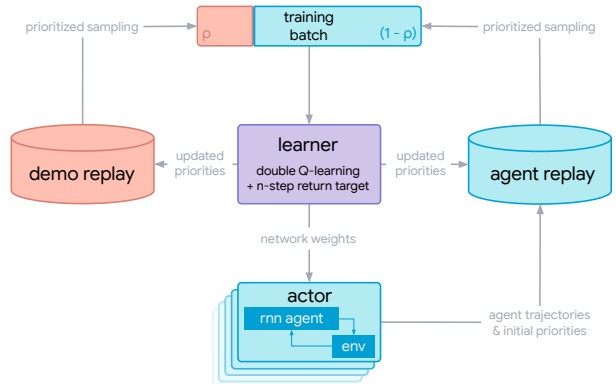

**Figure 1:** The R2D3 distributed system diagram. The learner samples batches that are a mixture of demonstrations and the experiences the agent generates by interacting with the environment over the course of training. The ratio between demos and agent experiences is a key hyper-parameter which must be carefully tuned to achieve good performance.

*agent replay* buffer, where experience from all actors is aggregated and globally prioritized (Schaul et al., 2016; Horgan et al., 2018) using a mixture of max and mean of the TD-errors with priority exponent $\eta = 1.0$ as in Kapturowski et al. (2018). The actors periodically request the latest network weights from the learner process in order to update their behavior.

In addition to the agent replay, we maintain a second *demo replay* buffer, which is populated with expert demonstrations of the task to be solved. Expert trajectories are also prioritized using the scheme of Kapturowski et al. (2018). Maintaining separate replay buffers for agent experience and expert demonstrations allows us to prioritize the sampling of agent and expert data separately.

The learner process samples batches of data from both the agent and demo replay buffers simultaneously. A hyperparameter $\rho$, the *demo ratio*, controls the proportion of data coming from expert demonstrations versus from the agent's own experience. The demo ratio is implemented at a batch level by randomly choosing whether to sample from the expert replay buffer independently for each element with probability $\rho$. Using a stochastic demo ratio in this way allows us to target demo ratios that are smaller than the batch size, which we found to be very important for good performance. The objective optimized by the learner uses of $n$-step, double Q-learning (with $n = 5$) and a dueling architecture (Wang et al., 2016; Hessel et al., 2018). In addition to performing network updates, the learner is also responsible for pushing updated priorities back to the replay buffers.

In each replay buffer, we store fixed-length ($m = 80$) sequences of $(s, a, r)$ tuples where adjacent sequences overlap by 40 time-steps. The sequences never cross episode boundaries. Given a single batch of trajectories we unroll both online and target networks (Mnih et al., 2015) on the same sequence of states to generate value estimates with the recurrent state initialized to zero. Proper initialization of the recurrent state would require always replaying episodes from the beginning, which would add significant complexity to our implementation. As an approximation of this we treat the first 40 steps of each sequence as a burn-in phase, and apply the training objective to the final 40 steps only. An alternative approximation would be to store stale recurrent states in replay, but we did not find this to improve performance over zero initialization with burn-in.

## 2.1 R2D3 Agent

In this section, we provide the pseudocode for the R2D3. First, the agent has a single **learner** process which samples from both demonstration and agent buffers in order to update its policy parameters, the pseudocode of the R2D3 learner can be found in Algorithm 1.

---
**Algorithm 1** Learner

**Inputs:** replay of expert demonstrations $\mathcal{D}$, replay of agent experiences $\mathcal{R}$, batch size $B$, sequence length $m$, and number of actors $A$.
Initialize policy weights $\theta$.
Initialize target policy weights $\theta' \leftarrow \theta$.
Launch $A$ actors and replicate policy weights $\theta$ to each actor.
**for** $n_{\text{steps}}$ **do**
    Sample transition sequences $(s_{t:t+m}, a_{t:t+m}, r_{t:t+m})$ from replay $\mathcal{D}$ with probability $\rho$ or from replay $\mathcal{R}$ with probability $(1 - \rho)$, to construct a mini-batch of size $B$.
    Calculate loss using target network.
    Perform a gradient descent step to update $\theta$.
    If $t \bmod t_{target} = 0$, update the target policy weights $\theta' \leftarrow \theta$.
    If $t \bmod t_{actor} = 0$, replicate policy weights to the actors.
**end for**

---

The R2D3 agent has $A$ parallel **actor** processes which interact with a copy of the environment in order to obtain data which is then inserted into the agent buffer. The agents periodically update their parameters to match those being updated on the learner. The pseudocode for the actors is provided in Algorithm 2.

---
**Algorithm 2** Actor

**repeat**
    Sample action from behavior policy $a \leftarrow \pi(s)$
    Execute $a$ and observe $s'$ and $r$
    Store $(s, a, s', r)$ in $\mathcal{R}$
**until** learner finishes.

---

## 3 Background

Exploration remains one of the most fundamental challenges for reinforcement learning. So-called "hard-exploration" domains are those in which rewards are sparse, and optimal solutions typically have

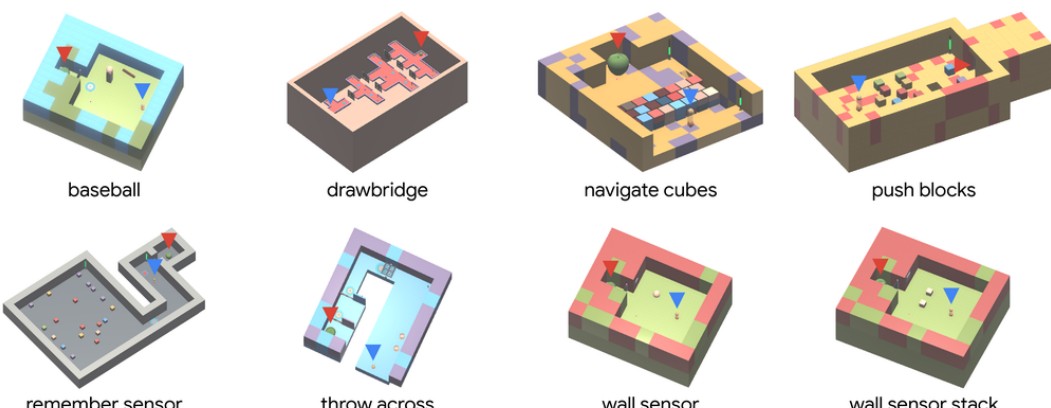

**Figure 2:** Hard-Eight task suite. In each task an agent (▼) must interact with objects in its environment in order to gain access to a large apple (▼) that provides reward. The 3D environment is also procedurally generated so that every episode the state of the world including object shapes, colors, and positions is different. From the point of view of the agent the environment is partially observed. Because it may take hundreds of low-level actions to collect an apple the reward is sparse which makes exploration difficult.

long and sparsely-rewarded trajectories. Hard-exploration domains may also have many distracting dead ends that the agent may not be able to recover from once it gets into a certain state. In recent years, the most notable such domains are Atari environments, including *Montezuma's Revenge* and *Pitfall* (Bellemare et al., 2013). These domains are particularly tricky for classical RL algorithms because even finding a single non-zero reward to bootstrap from is incredibly challenging.

A common technique used to address the difficulty of exploration is to encourage the agent to visit under-explored areas of the state-space (Schmidhuber, 1991). Such techniques are commonly known as intrinsic motivation (Chentanez et al., 2005) or count-based exploration (Bellemare et al., 2016). However, these approaches do not scale well as the state space grows, as they still require exhaustive search in sparse reward environments. Additionally, recent empirical results suggest that these methods do not consistently outperform $\epsilon$-greedy exploration (Taïga et al., 2019). The difficulty of exploration is also a consequence of the current inability of our agents to abstract the world and learn scalable, causal models with explanatory power. Instead they often use low-level features or handcrafted heuristics and lack the generalization power necessary to work in a more abstract space. Hints can be provided to the agent which bias it towards promising regions of the state space either via reward-shaping (Ng et al., 1999) or by introducing a sequence of curriculum tasks (Bengio et al., 2009; Graves et al., 2017). However, these approaches can be difficult to specify and, in the case of reward shaping, often lead to unexpected behavior where the agent learns to exploit the modified rewards.

Another hallmark of hard-exploration benchmarks is that they tend to be fully-observable and exhibit little variation between episodes. Nevertheless, techniques like random no-ops and "sticky actions" have been proposed to artificially increase episode variance in Atari (Machado et al., 2018), an alternative is to instead consider domains with inherent variability. Other recent work on the *Obstacle Tower* challenge domain (Juliani et al., 2019) is similar to our task suite in this regard. Reliance on determinism of the environment is one of the chief criticisms of imitation leveled by Juliani (2018), who offers a valuable critique on Aytar et al. (2018), Ecoffet et al. (2019) and Salimans and Chen (2018a). In contrast, our approach is able to solve tasks with substantial per-episode variability.

GAIL (Ho and Ermon, 2016) is another imitation learning method, however standard GAIL does not work in the following settings: 1) POMDPs (Gangwani et al., 2019; Żołna et al., 2019), 2) from pixels (Li et al., 2017; Reed et al., 2018), 3) off policy (Kostrikov et al., 2018) and 4) with variable initial conditions (Zolna et al., 2019). Our setting combines all of these, so we leave extending GAIL to this combined setting for future work.

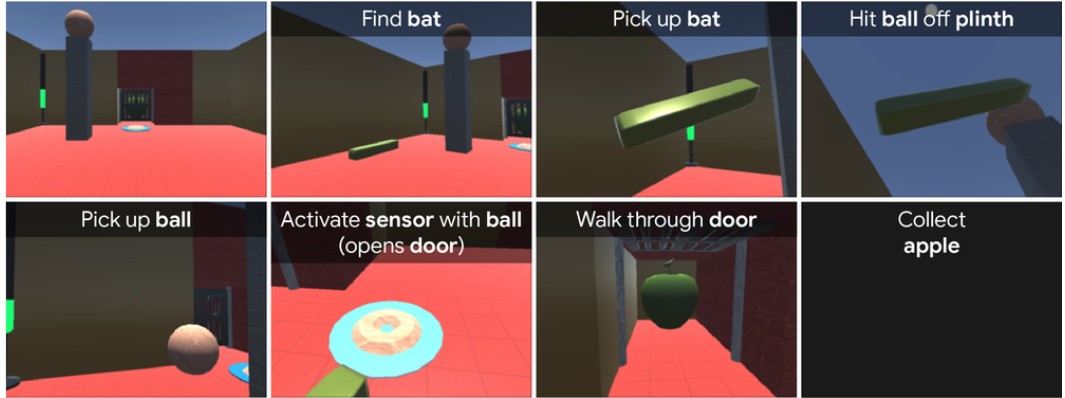

**Figure 3:** High-level steps necessary to solve the Baseball task. Each step in this sequence must be completed in order, and must be implemented by the agent as a sequence of low level actions (no option structure is available to the agent). The necessity of completing such a long sequence of high level steps makes it unlikely that the task will ever be solved by random exploration. Note that each step involves interaction with physical objects, shown in bold.

## 4 HARD-EIGHT TASK SUITE

To address the difficulty of hard exploration in partially observable problems with highly variable initital conditions we introduce a collection of eight tasks, which exhibit these properties. Due to the generated nature of these tasks and the rich form of interaction between the agent and environment, we see greatly increased levels of variability between episodes. From the perspective of the learning process, these tasks are particularly interesting because just memorizing an open loop sequence of actions is unlikely to achieve even partial success on a new episode. The nature of interaction with the environment combined with a limited field of view also necessitates the use of memory in the agent.

All of the tasks in the Hard-Eight task suite share important common properties that make them hard exploration problems. First, each task emits **sparse rewards**—in all but one task the only positive instantaneous reward obtained also ends the episode. The visual observations in each task are also first-person and thus the state of the world is only ever **partially observed**. Several of the tasks are constructed to ensure that that it is not possible to observe all task relevant information simultaneously.

Finally, each task is subject to a **highly variable initial conditions**. This is accomplished by including several procedural elements, including colors, shapes and configurations of task relevant objects. The procedural generation ensures that simply copying the actions from a demonstration is not sufficient for successful execution, which is a sharp contrast to the the case of Atari (Pohlen et al., 2018). A more detailed discussion of these aspects can be found in Appendix A and videos of agents and humans performing these tasks can be found at `https://bit.ly/2mAAUgg`.

Each task makes use of a **standardized avatar** with a first-person view of the environment, controlled by the same discretized action space consisting of 46 discrete actions. In all tasks the agent is rewarded for collecting apples and often this is the only reward obtained before the episode ends. A depiction of each task is shown in Figure 2. A description of the procedural elements and filmstrip of a successful episode for each task is provided in Appendix A.

Each of these tasks requires the agent to complete a sequence of high-level steps to complete the task. An example from the task suite is shown in Figure 3. The agent must: find the bat, pick up the bat, knock the ball off the plinth, pick up the ball, activate the sensor with the ball (opening the door), walk through the door, and collect the large apple.

We are hoping that our release of the Hard-Eight tasks [1] will enable machine learning researchers to try imitation learning or inverse reinforcement learning algorithms on more complicated tasks.

---

[1] The link for the tasks and the data can be found at `deepmind.com/r2d3`, once they are officially released.

## 5 BASELINES

In this section we discuss the baselines and ablations we use to compare against our R2D3 agent in the experiments. We compare to Behavior Cloning (a common baseline for learning from demonstrations) as well as two ablations of our method which individually remove either recurrence or demonstrations from R2D3. The two ablations correspond to two different state of the art methods from the literature.

**Behavior Cloning**  BC is a simple and common baseline method for learning policies from demonstrations (Pomerleau, 1989; Rahmatizadeh et al., 2018). This algorithm corresponds to a supervised learning approach to imitation learning which uses only expert trajectories as its training dataset to fit a parameterized policy mapping states to actions. For discrete actions this corresponds to a classification task, which we fit using the cross-entropy loss. If the rewards of trajectories in the training dataset are consistently high, BC is known to outperform recent batch-RL methods (Fujimoto et al., 2018). To enable fair comparison we trained our BC agent using the same recurrent neural network architecture that we used for our R2D3 algorithm (see Figure 4).

**No Demonstrations**  The first ablation we consider is to remove demonstrations from R2D3. This corresponds to setting the demo ratio (see Figure 1) to $\rho = 0$. This special case of R2D3 corresponds exactly to the R2D2 agent of Kapturowski et al. (2018), which itself extends DQN (Mnih et al., 2015) to partially observed environments by combining it with recurrence and the distributed training architecture of Ape-X DQN (Horgan et al., 2018). This ablation is itself state of the art on Atari-57 and DMLab-30, making it an extremely strong baseline.

**No Recurrence**  The second ablation we consider is to replace the recurrent value function of R2D3 with a feed-forward reactive network. We do this separately from the no demonstrations ablation, leaving the full system in Figure 1 in tact, with only the structure of the network changed. If we further fix the demo ratio to $\rho = 0.25$ then this ablation corresponds to the DQfD agent of Hester et al. (2018), which is competitive on hard-exploration Atari environments such as Montezuma's Revenge. However, we do not restrict ourselves to $\rho = 0.25$, and instead optimize over the demo ratio for the ablation as well as for our main agent.

## 6 EXPERIMENTS

We evaluate the performance of our R2D3 agent alongside state-of-the-art deep RL baselines. As discussed in Section 5, we compare our R2D3 agent to BC (standard LfD baseline) R2D2 (off-policy SOTA), DQfD (LfD SOTA). We use our own implementations for all agents, and we plan to release code for all agents including R2D3.

For each task in the Hard-Eight suite, we trained R2D3, R2D2, and DQfD using 256 $\epsilon$-greedy CPU-based actors and a single GPU-based learner process. Following Horgan et al. (2018), the $i$-th actor was assigned a distinct noise parameter $\epsilon_i \in [0.4^8, 0.4]$ where each $\epsilon_i$ is regularly spaced in $\log_{0.4}$ space. For each of the algorithms their common hyperparameters were held fixed. Additionally, for R2D3 and DQfD the demo ratio was varied to study its effect. For BC we also varied the learning rate independently in a vain attempt to find a successful agent.

All agents act in the environment with an action-repeat factor of 2, i.e. the actions received by the environment are repeated twice before passing the observation to the agent. Using an action repeat of 4 is common in other domains like Atari (Bellemare et al., 2012; Mnih et al., 2015); however, we found that using an action repeat of 4 made the Hard-Eight tasks too difficult for our demonstrators. Using an action repeat of 2 allowed us to strike a compromise between ease of demonstration (high action repeats prohibiting smooth and intuitive motion) and ease of learning for the agents (low action repeats increase the number of steps required to complete the task).

Figure 4 illustrates the neural network architecture of the different agents. As much as possible we use the same network architecture across all agents, deviating only for DQfD, where the recurrent head is replaced with an equally sized feed-forward layer. We briefly outline the training setup below, and give an explicit enumeration of the hyperparameters in Appendix B.

For R2D3, R2D2 and DQfD we use the Adam optimizer (Kingma and Ba, 2014) with a fixed learning rate of $2 \times 10^{-4}$. We use hyperparameters that are shown to work well for similar environments. We use distributed training with 256 parallel actors, trained for at least 10 billion actor steps for all tasks.

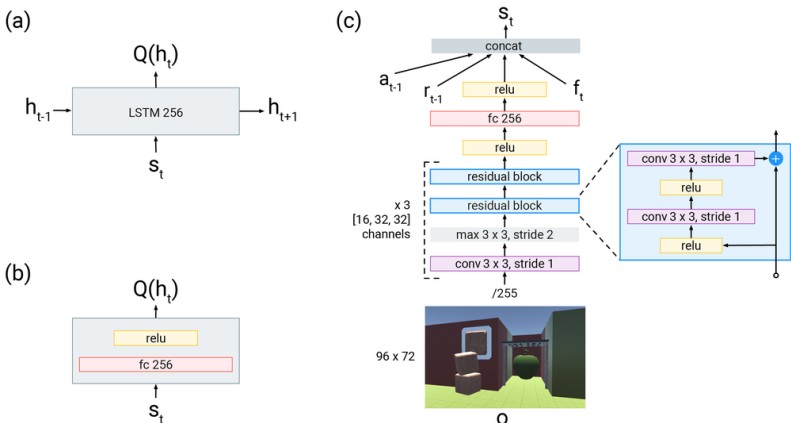

**Figure 4: (a)** Recurrent head used by R2D3 agents. **(b)** Feedforward head used by the DQfD agent. Heads in both a) and b) are used to compute the Q values. **(c)** Architecture used to compute the input feature representations. Frames of size 96x72 are fed into a ResNet, the output is then augmented by concatenating the previous action $a_{t-1}$, previous reward $r_{t-1}$, and other proprioceptive features $f_t$, such as accelerations, whether the avatar hand is holding an object, and the hand's relative distance to the avatar.

For the BC agent the training regime is slightly different, since this agent does not interact with the environment during training. For BC we also use the Adam optimizer but we additionally perform a hyperparameter sweep over learning rates $\{10^{-5}, 10^{-4}, 10^{-3}\}$. Since there is no notion of actor steps in BC we trained for 500k learner steps instead.

During the course of training, an evaluator process periodically queries the learner process for the latest network weights and runs the resulting policy on an episode, logging both the final return and the total number of steps (actor or learner steps, as appropriate) performed at the time the of evaluation.

We collected a total of 100 demonstrations for each task spread across three different experts (each expert contributed roughly one third of the demonstrations for each task). Demonstrations for the tasks were collected using keyboard and mouse controls mapped to the agent's exact action space, which was necessary to enable both behaviour cloning and learning from demonstrations. We show statistics related to the human demonstration data which we collected from three experts in Table 1.

## 6.1 LEARNING THE HARD-EIGHT TASKS

In Figure 5, we report the return against the number of actor steps, averaged over five random initializations. We find that none of the baselines succeed in any of the eight environments. Meanwhile, R2D3 learns six out of the eight tasks, and reaches or exceeds human performance in four of them. The fact that R2D3 learns at all in this setting with only 100 demonstrations per task demonstrates the ability of the agent to make very efficient use of the demonstrations. This is in contrast to BC and DQfD which use the same demonstrations, and both fail to learn a single task from the suite.

All methods, including R2D3, fail to solve two of the tasks: Remember Sensor and Throw Across. These are the two tasks in the suite that are most demanding in terms of memory requirements for the agent, and it is possible that our zero-initialization with burn-in strategy for handling LSTM states in replay does not give R2D3 sufficient context to complete these tasks successfully. Future work should explore the better handling of recurrent states as a possible avenue towards success on these tasks. R2D3, BC, and DQfD receive some negative returns on Remember Sensor, which indicates that the agents navigate down the hallway and walks over penalty sensors.

R2D3 performed better than our average human demonstrator on Baseball, Drawbridge, Navigate Cubes and the Wall Sensor tasks. The behavior on Wall Sensor Stack in particular is quite interesting. On this task R2D3 found a completely different strategy than the human demonstrators by exploiting a bug in the implementation of the environment. The intended strategy for this task is to stack two blocks on top of each other so that one of them can remain in contact with a wall mounted sensor,

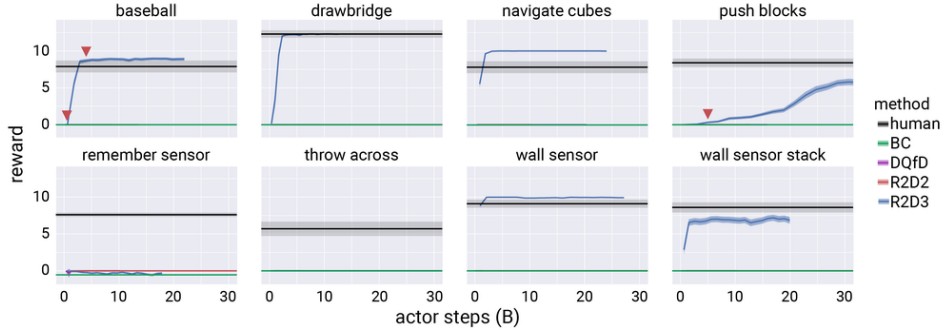

**Figure 5:** Reward vs actor steps curves for R2D3 and baselines on the Hard-Eight task suite. The curves are computed as the mean performance for the same agent across 5 different seeds per task. Error regions show the 95% confidence interval for the mean reward across seeds. Several curves overlap exactly at zero reward for the full range of the plots. R2D3 can perform human-level or better on Baseball, Drawbridge, Navigate Cubes and Wall Sensor. R2D2 could not get any positive rewards on any of the tasks. DQfD and BC agents occasionally see rewards on Drawbridge and Navigate Cubes tasks, but this happens rarely enough that the effect is not visible in the plots. Indicators (▼) mark analysis points in Section 6.3.

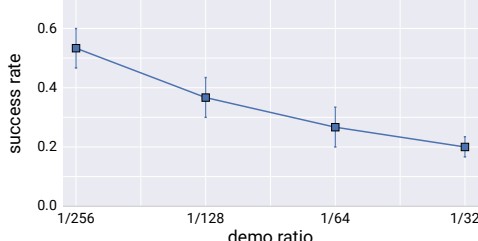

Figure 6 | Success rate (see main text) for R2D3 across all tasks with at least one successful seed, as a function of demo ratio. The square markers for each demo ratio denote the mean success rate, and the error bars show a bootstrapped estimate of the $[25, 75]$ percentile interval for the mean estimate. The lower demo ratios consistently outperform the higher demo ratios across the suite of tasks.

| Task Name | Reward | Episode Len. |
|---|---|---|
| Baseball | $7.8 \pm 4.1$ | $492 \pm 121$ |
| Drawbridge | $12.3 \pm 2.5$ | $641 \pm 137$ |
| Navigate Cubes | $7.9 \pm 4.1$ | $638 \pm 185$ |
| Push Blocks | $9.1 \pm 2.9$ | $683 \pm 270$ |
| Remember Sensor | $7.7 \pm 1.4$ | $853 \pm 188$ |
| Throw Across | $5.4 \pm 4.9$ | $464 \pm 172$ |
| Wall Sensor | $9.1 \pm 2.8$ | $280 \pm 87$ |
| Wall Sensor Stack | $8.6 \pm 3.5$ | $521 \pm 107$ |

**Table 1** | Human demonstration statistics. We collected 100 demos for each tasks from three human demonstrators. We report mean lengths (in number of frames) and rewards of the episodes along with the standard deviations for each task.

and this is the strategy employed by the demonstrators. However, due to a bug in the environment the strategy learned by R2D3 was to trick the sensor into remaining active even when it is not in contact with the key by pressing the key against it in a precise way.

In light of the uniform failure of our baselines to learn on the Hard-Eight suite we made several attempts at training other models on the task suite; however, these attempts were all unsuccessful. For example, we tried adding randomized prior functions (Osband et al., 2018) to R2D2, but this approach was still unable to obtain reward on any of the Hard-Eight tasks. We also trained an IMPALA agent with pixel control (Jaderberg et al., 2016) as auxiliary reward to help with exploration, but this approach also failed to learn on any of the tasks we attempted. We omit these results from Figure 5, only keeping the most relevant baselines.

## 6.2 EFFECT OF THE DEMO RATIO

In our experiments on Hard-Eight tasks (see Figure 5), we did a hyperparameter search and chose the best hyperparameters for each method independently. In this section, we look more closely at how the demo ratio ($\rho$) affects learning in R2D3. To do this we look at how the success rate of R2D3 across the entire Hard-Eight task suite varies as a function of the demo ratio.

The goal of each task in the Hard-Eight suite is to collect a large apple, which ends the episode and gives a large reward. We consider an *episode* successful if the large apple is collected. An agent that executes many episodes in the environment will either succeed or fail at each one. We consider

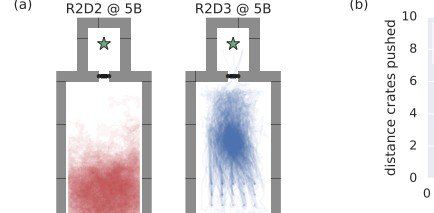 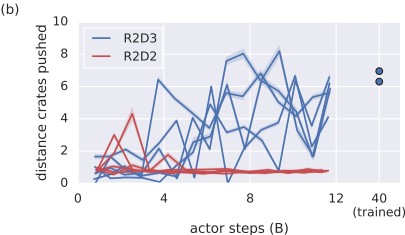 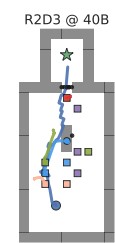

**Figure 7:** Guided exploration behavior in the Push Blocks task. **(a)** Spatial pattern of exploration behavior at ~5B actor steps (reward-driven learning kicks off for R2D3 only after ~20B steps). Overlay of agent's trajectories over 200 episodes. Blocks and sensors are not shown for clarity. R2D2 appears to follow a random walk. R2D3 concentrates on a particular spatial region. **(b)** Interactions between the agent and blocks during the first 12B steps. Each line shows a different random seed. R2D2 rarely pushes the blocks. **(c)** Example trajectory of R2D3 after training, the agent pushes the blue block onto the blue sensor, then collects the apple (green star).

an *agent* successful if, after training, at least 75% of its final 25 episodes are successful. Finally, an individual agent with a fixed set of hyperparameters may still succeed or fail depending on the randomness in the environment and the initialization of the agent.

We train several R2D3 agents on each tractable task[2] in the Hard-Eight suite, varying only the demo ratio while keeping other hyperparameters fixed at the values used for the learning experiment. We consider four different demo ratios across six tasks, with five seeds for each task (120 trained agents). Figure 6 shows estimates of the success rate for the R2D3 algorithm for each different demo ratio, aggregated across all tasks. We observe that tuning the demo ratio has a strong effect on the success rate across the task suite, and that the best demo ratio is quite small. See Appendix C.3 for further results.

### 6.3 GUIDED EXPLORATION BY DEMONSTRATION

The typical strategy for exploration in RL is to either use a stochastic policy and sample actions, or to use a deterministic policy and take random actions some small $\epsilon$ fraction of the time. Given sufficient time both of these approaches will in theory cover the space of possible behaviors, but in practice the amount of time required to achieve this coverage can be prohibitively long. In this experiment, we compare the behavior of R2D3 to the behavior of R2D2 (which is equivalent to R2D3 without demonstrations) on two of the tasks from the Hard-Eight suite. Even very early in training (well before R2D3 is able to reliably complete the tasks) we see many more task-relevant actions from R2D3 than from R2D2, suggesting that the effect of demonstrations is to bias R2D3 towards exploring relevant parts of the environment.

In Figure 7 we begin by examining the Push Blocks tasks. The task here is to push a particular block onto a sensor to give access to a large apple, and we examine the behavior of both R2D3 and R2D2 after 5B steps, which is long before R2D3 begins to solve the task with any regularity (see Figure 5). Looking at the distribution of spatial locations for the agents it is clear that R2D2 essentially diffuses randomly around the room, while R2D3 spends much more time in task-relevant parts of the environment (e.g. away from the walls). We also record the total distance traveled by the moveable blocks in the room, and find that R2D3 tends to move the blocks significantly more often than R2D2, even before it has learned to solve the task.

## 7 CONCLUSION

In this paper, we introduced the R2D3 agent, which is designed to make efficient use of demonstrations to learn in partially observable environments with sparse rewards and highly variable initial conditions. We showed through several experiments on eight very difficult tasks that our approach is able to outperform multiple state of the art baselines, two of which are themselves ablations of R2D3.

---

[2]We exclude Remember Sensor and Throw Across from this analysis, since we saw no successful seeds for either of these tasks.

We also identified a key parameter of our algorithm, the *demo ratio*, and showed that careful tuning of this parameter is critical to good performance. Interestingly we found that the optimal demo ratio is surprisingly small but non-zero, which suggests that there may be a risk of overfitting to the demonstrations at the cost of generalization. For future work, we could investigate how this optimal demo ratio changes with the total number of demonstrations and, more generally, the distribution of expert trajectories relative to the task variability.

We introduced the Hard-Eight suite of tasks and used them in all of our experiments. These tasks are specifically designed to be partially observable tasks with sparse rewards and highly variable initial conditions, making them an ideal testbed for showcasing the strengths of R2D3 in contrast to existing methods in the literature.

Our behavioral analysis showed that the mechanism R2D3 uses to efficiently extract information from expert demonstrations is to use them in a way that guides (or biases) the agent's own autonomous exploration of the environment. An in-depth analysis of agent behavior on the Hard-Eight task suite is a promising direction for understanding how different RL algorithms make selective use of information.

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

## A    HARD-EIGHT TASK SUITE DETAILS

**Sparse rewards**    All of the tasks emit sparse rewards, indeed in all but one task the only positive instantaneous reward obtained also ends the episode successfully. In other words, for standard RL algorithms to learn by bootstrapping, the actors must first solve the task inadvertently, and must do so with no intermediate signal to guide them.

**Partial observability**    Visual observations are all first-person, which means that some relevant features of the state of the world may be invisible to the agent simply because they are behind it or around a corner. Some tasks (e.g. Remember Sensor, are explicitly designed so that this is the case).

**Highly Variable Initial Conditions**    Many of the elements of the tasks are procedurally generated, which leads to significant variability between episodes of the same task. In particular, the starting position and orientation of the agent are randomized and similarly, where they are present, the shapes, colors, and textures of various objects are randomly sampled from a set of available such features. Therefore a single (or small number of) demonstration(s) is not sufficient to guide an agent to solve the task as it is in the case of DQfD on Atari (Pohlen et al., 2018).

**Observation specification**    All of the tasks provide the same observation space. In particular, a visual channel consisting of 96 by 72 RGB pixels, as well as accelerations of the avatar, force applied by the avatar hand on the object, whether if the avatar is holding anything or not, and the distance of a held object from the face of the avatar (zero when there is no held object).

**Action specification**    The action space consists of four displacement and four rotation actions (8), duplicated for coarse and fine-grained movement (16) as well as for movement with and without grasping (32). The avatar also has an invisible "hand" which can be used to manipulate objects in the environment. The location of the hand is controlled by the avatar gaze direction, plus an additional two actions that control the distance of the hand from the body (34). A grasped object can be manipulated by six rotation actions (two for each rotational degree of freedom; 40) as well as four additional actions controlling the distance of the hand from the body at coarse and fine speed (44). Finally there is an independent grasp action (to hold an object without moving), and a no-op action (total 46). Compared to course actions, fine-grained actions result in slower movements, allowing the agent to perform careful manipulations.

### A.1    INDIVIDUAL TASK DETAILS

This section gives addition details on each task in our suite including a sequence frames from a successful task execution (performed by a human) and a list of the procedural elements randomized per episode. Videos of agents and humans performing these tasks can be found at `https://bit.ly/2mAAUgg`.

**Baseball**

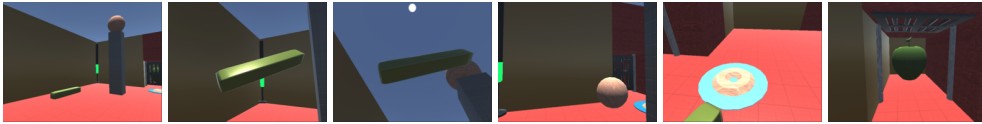

The agent spawns in a small room with a sensor and a key object resting high atop a plinth. The agent must find a stick and use it to knock the key object of the plinth in order to activate the sensor. Activating the sensor opens a door to an adjoining room with a large apple which ends the episode.

Procedural elements

- Initial position and orientation of the agent
- Wall, floor and object materials and colors
- Initial position of the stick
- Position of plinth

**Drawbridge**

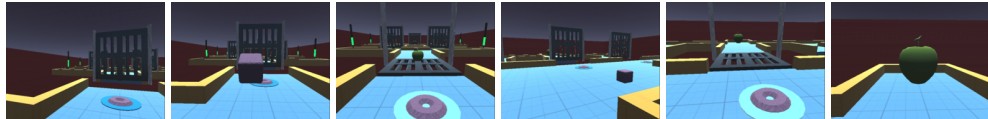

The agent spawns at one end of a network of branching platforms separated by drawbridges, which can be activated by touching a key object to a sensor. Activating a drawbridge with a key object destroys the key. Each platform is connected to several drawbridges, but has only one key object available. Some paths through the level have small apples which give reward. The agent must choose the most rewarding path through the level to obtain a large apple at the end which ends the episode.

Procedural elements

- Initial position and orientation of the agent

- Wall, floor, ceiling and object materials and colors

- Positions of the small apples throughout the network of ledges

**Navigate Cubes**

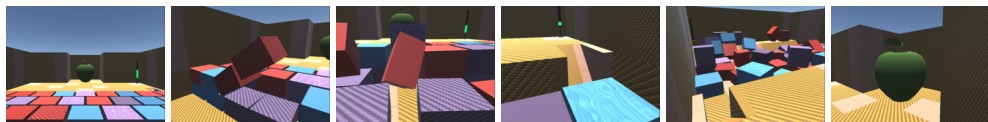

The agent spawns on one side of a large room. On the other side of the room on a raised platform there is a large apple which ends the episode. Across the center of the room there is a wall of movable blocks. The agent must dig through the wall of blocks and find a ramp onto the goal platform in order to collect the large apple.

Procedural elements

- Initial position and orientation of the agent

- Wall, floor and object materials and colors

**Push Blocks**

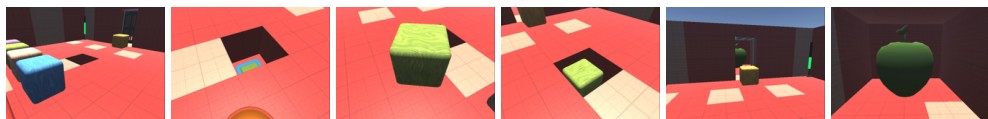

The agent spawns in a medium sized room with a recessed sensor in the floor. There are several objects in the room that can be pushed but not lifted. The agent must push a block whose color matches the sensor into the recess in order to open a door to an adjoining room which contains a large apple which ends the episode. Pushing a wrong object into the recess makes the level impossible to complete.

Procedural elements

- Initial position and orientation of the agent

- Wall, floor, object materials and colors

- Positions of the objects

- Sensor required color

**Remember Sensor**

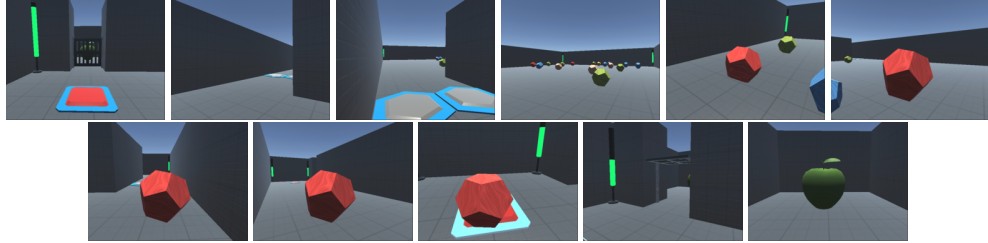

The agent spawns near a sensor of a random color. The agent must travel down a long hallway to a room full of blocks and select one that matches the color of the sensor. Bringing the correct block back to the sensor allows access to a large apple which ends the episode. In addition to being far away, traveling between the hallway and the block room requires the agent to cross penalty sensors which incurs a small negative reward.

Procedural elements

- Initial position and orientation of the agent

- Sensor required color

- Number of objects in the block room

- Position of objects in the block room

- Shape and material of the objects in the block room

**Throw Across**

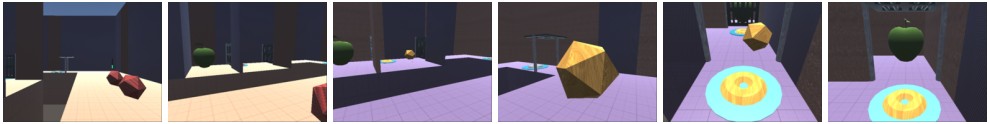

The agent spawns in a U shaped room with empty space between the legs of the U. There are two key objects near the agent spawn point. The agent must throw one of the key objects across the void, and carry the other around the bottom of the U. Both key objects are needed to open two locked doors which then give access to a large apple which ends the episode.

Procedural elements

- Initial position and orientation of the agent

- Wall, floor and object materials and colors

- Color and material of the sensors

- Initial positions of the two key objects

**Wall Sensor**

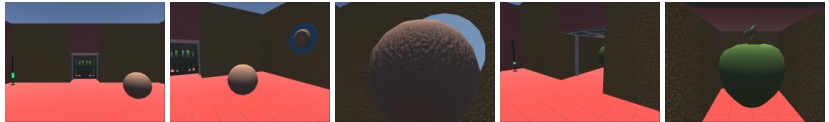

The agent spawns in a small room with a wall mounted sensor and a key object. The agent must pick up the key and touch it to the sensor which opens a door. In the adjoining room there is a large apple which ends the episode.

Procedural elements

- Initial position and orientation of the agent

- Position of the sensor

- Position of the key object

**Wall Sensor Stack**

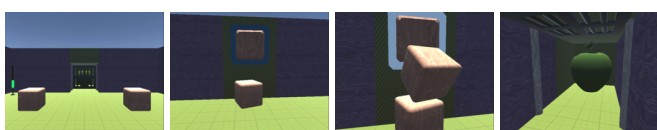

The agent spawns in a small room with a wall mounted sensor and two key objects. This time one of key objects must be in constant contact with the sensor in in order for the door to remain open. The agent must stack the two objects so one can rest against the sensor, allowing the agent to pass through to an adjoining room with a large apple which ends the episode.

Procedural elements

- Initial position and orientation of the agent

- Wall, floor and object materials and colors

- Initial positions of both key objects

- Position of the sensor

## B  HYPER-PARAMETERS

In Table 2, we report the shared set of hyper-parameters across different models and tasks.

| Hyperparameters | Values |
| --- | --- |
| **Network** | See Figure 4 |
| **Environment** | |
| Image height | 72 |
| Image width | 96 |
| Color | RGB |
| Action repeats | 2 |
| Observation spec | See section A |
| Action spec | See section A |
| **Learner** | |
| Learning rate | 2e-4 |
| Optimizer | Adam (Kingma and Ba, 2014) |
| Global norm gradient clipping | True |
| Discount factor ($\gamma$) | 0.997 |
| Batch size ($B$) | 32 |
| Target update period ($t_{target}$) | 400 |
| Actor update period ($t_{actor}$) | 200 |
| Prioritized sampling | True |
| Sequence length ($m$) | 80 |
| Burn in length | 40 |
| Asymmetric reward clipping | True |
| Number of actors ($A$) | 256 |
| Max replay capacity | 500000 |
| Min replay capacity | 25000 |

**Table 2:** Hyper-parameters used for all experiments.

## C  EXPERIMENTS

### C.1  SURPASSING THE EXPERTS

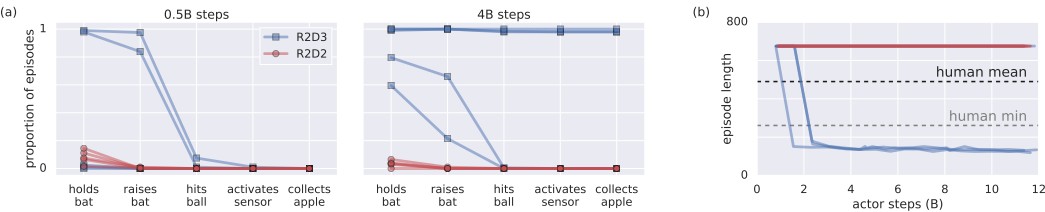

**Figure 8:** Guided exploration behavior in the Baseball task. **(a)** Sub-behaviors expressed by five R2D2 and five R2D3 agents after 0.5B steps of training (left) and 4B steps of training (right). Each point is estimated from 200 episodes. At 0.5B steps, none of the agents received any reward over the 200 evaluation episodes, while at 4B steps, three of the R2D3 agents received reward on almost every episode. Even when the R2D3 agents are not receiving reward, they are expressing some of the necessary behaviors provided through human demonstrations. **(b)** R2D3 agents eventually surpass human performance. The 3 of 5 R2D3 agents shown in (a) which start obtaining rewards continue to bootstrap towards more efficient policies than humans.

An important property of R2D3 is that although the agents are trained from demonstrations, the behaviors they achieve are able to surpass the skill of the demonstrations they were trained from. This can be seen quantitatively from reward curves in Figure 5, where the R2D3 agent surpasses the

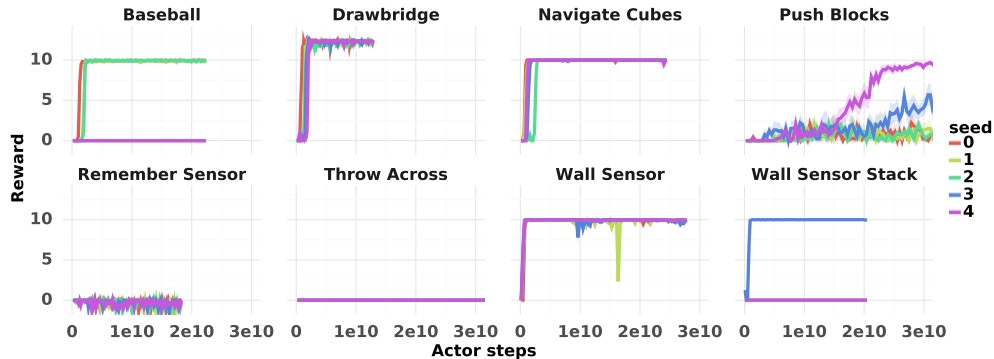

**Figure 9:** We show the rewards of the R2D3 agent on different tasks for each seed separately.

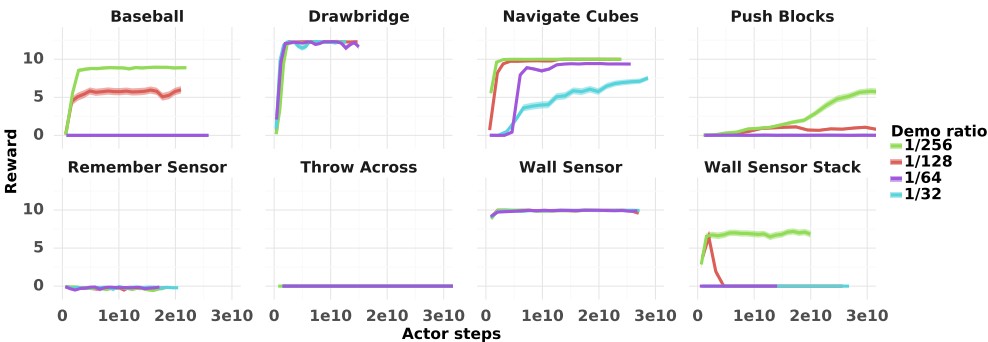

**Figure 10:** R2D3 learning curves with varying demo ratios for all tasks.

human baseline performance on four of the eight tasks (e.g. Baseball, Navigate Cubes, Wall Sensor and Wall Sensor Stack).

In some of these cases the improved score is simply a matter of executing the optimal strategy more fluently than the demonstrators. For example, this is the case in the Baseball task, where the human demonstrators are handicapped by the fact that the human interface to the agent action space makes it awkward to rotate a held object. This makes picking up the stick and orienting it properly to knock the ball off the plinth into a tricky task for humans, but the agents are able to refine their behavior to be much more efficient (see Figure 8c).

The behavior on Wall Sensor is especially interesting, however in this case the agents find a completely different strategy than the human demonstrators by exploiting a bug in the implementation of the environment. The intended strategy for this task is to stack two blocks on top of each other so that one of them can remain in contact with a wall mounted sensor, and this is the strategy employed by the demonstrators. However, due to a bug in the environment it is also possible to trick the sensor into remaining active even when it is not in contact with the key by pressing the key against it in a precise way. The R2D3 agents are able to discover this bug and exploit it, resulting in superhuman scores on this task even though this strategy is not present in the demonstrations.

## C.2 ADDITIONAL EXPERIMENTS

We also ran a few additional experiments to get more information about the tasks we did not solve, or solved incorrectly. Videos for these experiments are available at `https://bit.ly/2mAAUgg`.

**Remember Sensor**   This task requires a long memory, and also has the longest episodes length of any task in the Hard Eight suite. In an attempt to mitigate these issues, we trained the agent using a higher action repeat which reduces the episode length, and used stale lstm states instead of zero lstm states which provides information from earlier in the episode. This allows R2D3 to learn policies that display reasonable behavior, retrieving a random block and bringing it back to the hallway. Using this method it can occasionally solve the task.

**Throw Across** The demonstrations collected for this task had a very low success rate of 54%. We attempted to compensate for this by collecting an additional 30 demos. When we trained R2D3 with all 130 demos all seeds solved the task.

**Wall Sensor Stack** The original Wall Sensor Stack environment had a bug that the R2D3 agent was able to exploit. We fixed the bug and verified the agent can learn the proper stacking behavior.

## C.3 ADDITION DETAILS FOR MAIN EXPERIMENTS

In Figure 9, we show the performance of the R2D3 agents for each seed separately. On task such as Drawbridge, Navigate Cubes and Wall Sensor, all seeds take off quite rapidly and they have very low variance for the rewards between different seeds. However, on Wall Sensor Stack task while one seed takes off quite rapidly, and the rest of them are just flat. In Figure 10, we elaborate on Figure 6. For Baseball, Navigate Cubes, Push Blocks, and Wall Sensor Stack, a demo ratio of 1/256 works best. On Drawbridge and Wall Sensor all demo ratios are similarly effective.

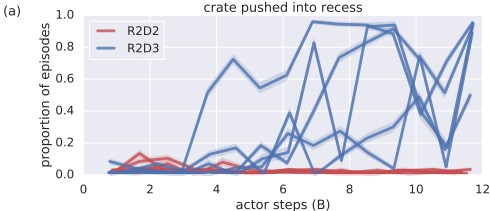 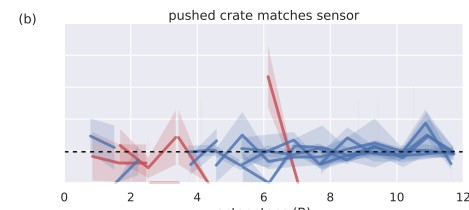

**Figure 11:** Further detail of guided exploration behavior in the Push Blocks task (as in Figure 7). **(a)** Proportion of episodes in which the agent pushes a crate into the recess during the initial 12B steps of training. **(b)** Proportion of episodes in which the crate pushed into the recess actually matches the sensor color. Data are only shown when crates are pushed into the recess on at least 5 out of 200 episodes. Dashed line shows the probability expected if a random crate was pushed into the recess. Thus, while (c) shows that by 12B steps the R2D3 agent may have reasonable success in pushing crates into the recess, it has not yet mastered the logic that the crate color must much the sensor color.

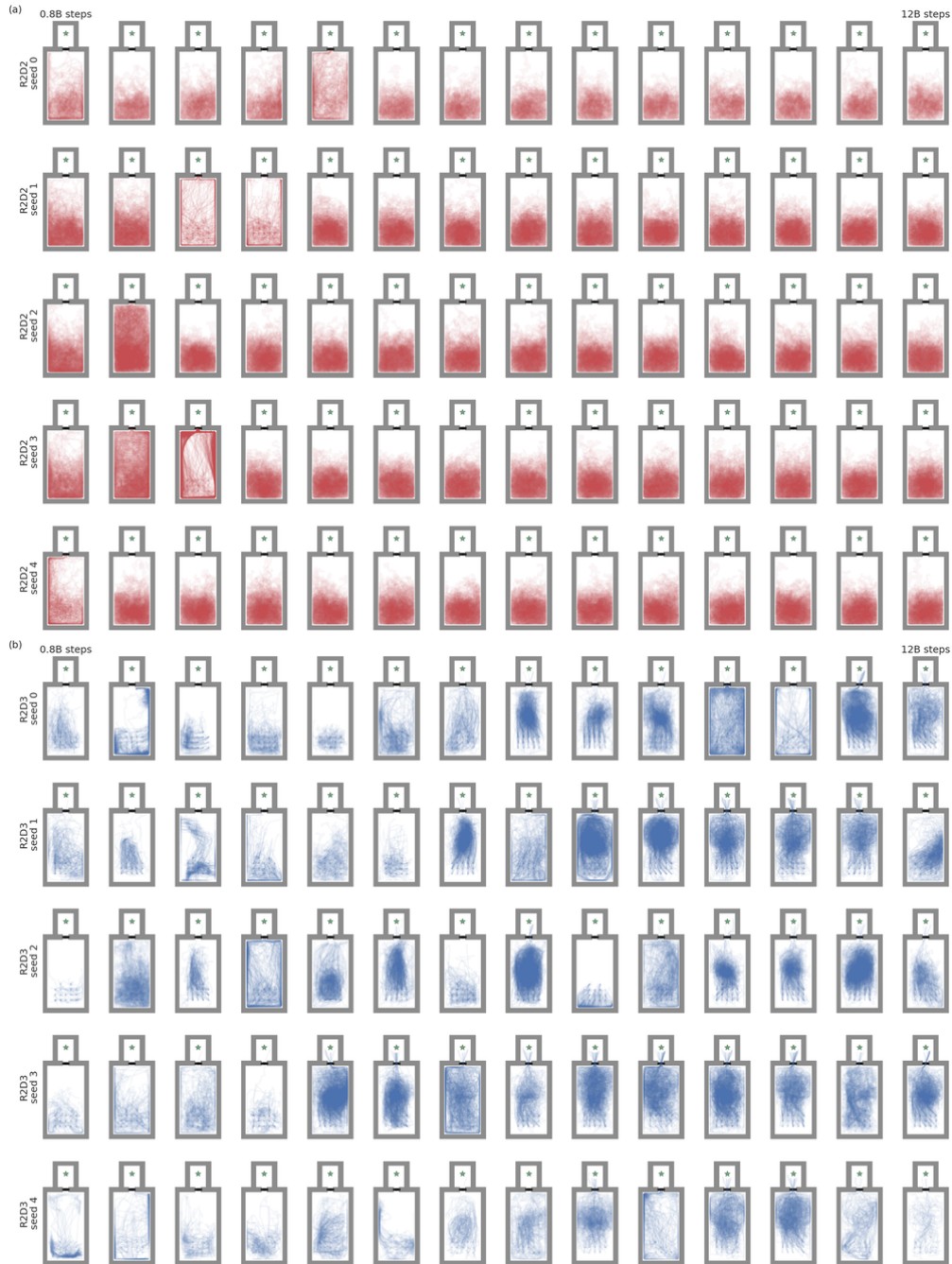

**Figure 12:** Further detail of guided exploration behavior in the Push Blocks task (as in Figure 7). **(a)** Spatial pattern of exploration behavior for the R2D2 agent over the course of ∼12B steps of training. Each row shows a different random seed; the number of training steps increases from the leftmost column to the rightmost column. There is little variation in how the policy manifests as explorative behavior across seeds and training time. **(b)** As in (a), for R2D3. Given demonstrations, the policies now show substantial variation across seeds and training time.

