# OpenReview forum: "Making Efficient Use of Demonstrations to Solve Hard Exploration Problems"
_ICLR.cc/2020/Conference — Accept (Poster)_

### Official Review · AnonReviewer2 · 2019-10-22
**Official Blind Review #2**

**Rating:** 6

**Review:**

Summary
-------------
The authors propose R2D3, and algorithm for learning from demonstrations in partially-observable environments with sparse rewards. The algorithm combines DQfD with recurrent networks to both leverage expert demonstrations and handle partial observability. Furthermore, the authors propose a suite of eight challenging tasks on which the proposed method is tested and compared to relevant baselines.

Comments
--------

Scaling RL agents to high-dimensional partially-observable domains with sparse rewards is a fundamental open problem and this work provides a nice contribution towards its solution. The paper is well-written and easy to read. The proposed methodology seems to be a simple combination of existing algorithms and (apologies if I am wrong) I did not see any particular challenge in its design. On the other hand, the hard-eight task suite is interesting and, if released, could be used as a benchmark by the whole community. The experiments seem quite convincing in proving the potential of the proposed method. Some comments/questions follow.

1. Section 2 presents the algorithm from a very high-level perspective. If space in the final version allows it, I would also suggest adding a more detailed pseudo-code to the main text, so that even a reader who is not completely familiar with the works this method builds upon could better understand and possibly implement the method.

2. Since the authors compare to behavioral cloning to prove the benefits over simple imitation methods, why not comparing to stronger baselines such as [1] or [2]?

3. The demo-ratio seems to be the key parameter to make this approach work (and the performance is proven very sensitive to its value). Instead of keeping it fixed across the entire learning process, have you tried to start with a high value and then decay according to a proper schedule? Intuitively, I would expect the benefits of expert demonstrations to be more valuable during the first learning episodes (where they make the agent explore better) and less during the successive phases (where the policy gets closer and closer to optimal).

4. The way recurrent states are handled with zero-initialization is probably one of the limitations and seems to play an important role in some experiments. Have you tried, at least in simpler domains, to replay whole episodes and see whether that helps?

[1] Ho, J., & Ermon, S. (2016). Generative adversarial imitation learning. In Advances in neural information processing systems (pp. 4565-4573).
[2] Finn, C., Levine, S., & Abbeel, P. (2016, June). Guided cost learning: Deep inverse optimal control via policy optimization. In International Conference on Machine Learning (pp. 49-58).

**Experience Assessment:**

I have read many papers in this area.

**Review Assessment: Checking Correctness Of Derivations And Theory:**

N/A

**Review Assessment: Checking Correctness Of Experiments:**

I assessed the sensibility of the experiments.

**Review Assessment: Thoroughness In Paper Reading:**

I read the paper at least twice and used my best judgement in assessing the paper.

---

> ### Author Response · Authors · 2019-11-13
> **Reply to Official Review #2**
>
> > On the other hand, the hard-eight task suite is interesting and, if released, could be used as a benchmark by the whole community.
>
> Thank you. We are working to make these environments available to the community in time for ICLR.
>
> > Section 2 presents the algorithm from a very high-level perspective. If space in the final version allows it, I would also suggest adding a more detailed pseudo-code to the main text, so that even a reader who is not completely familiar with the works this method builds upon could better understand and possibly implement the method.
>
> That is a great suggestion. We will add the pseudocode to the final version of our paper.
>
> > Since the authors compare to behavioral cloning to prove the benefits over simple imitation methods, why not comparing to stronger baselines such as [1] or [2]?
>
> [1] and [2] are both examples of Imitation Learning by Inverse Reinforcement Learning. These methods are very powerful, but we did not try them because there is a strong evidence in the literature that standard versions of these algorithms do not work in the following settings: 1) POMDPs [a], 2) from pixels [b, c], 3) off policy [d] and 4) with variable initial conditions [e]. Our setting combines all of these. Each IL by IRL extension cited above is nontrivial and combining them may present challenges which are beyond the scope of this work.
>
> We are hoping that our release of the Hard-Eight tasks will enable other researchers to try IL by IRL on more complicated tasks.
>
> [a] Learning Belief Representations for Imitation Learning in POMDPs. 2019.
> [b] InfoGAIL: Interpretable Imitation Learning from Visual Demonstrations. 2017.
> [c] Visual Imitation with a Minimal Adversary. 2018.
> [d] Discriminator-Actor-Critic: Addressing Sample Inefficiency and Reward Bias in Adversarial Imitation Learning. 2019.
> [e] Task-Relevant Adversarial Imitation Learning. 2019.
>
> >  The demo-ratio seems to be the key parameter to make this approach work (and the performance is proven very sensitive to its value). Instead of keeping it fixed across the entire learning process, have you tried to start with a high value and then decay according to a proper schedule? Intuitively, I would expect the benefits of expert demonstrations to be more valuable during the first learning episodes (where they make the agent explore better) and less during the successive phases (where the policy gets closer and closer to optimal).
>
> We agree with this proposal, annealing the demo ratio can be an interesting experiment to try. Our experiments are already compute intensive and doing hyperparameter search for the annealing hyperparameters would require even more compute. As a result, we decided to just fix the demo-ratio throughout the training. As a future work, it would be interesting to evaluate different annealing methods with R2D3.
>
> > The way recurrent states are handled with zero-initialization is probably one of the limitations and seems to play an important role in some experiments. Have you tried, at least in simpler domains, to replay whole episodes and see whether that helps?
>
> Good point. We have tried two variations: 1) replaying the whole episode as you described, and 2) using stale lstm states as described in R2D2. Both of these variations seem to help to some extent on the the hardest memory task, remember sensor [a], but they do not help on the other tasks. We didn't focus on these variations because they introduce additional complexity [b] and do not change performance on most tasks.
>
> [a] the agent seems more reward, but still fails to solve the task 100% of the time
> [b] 1) multi-gpu training to allow for full unrolls over the whole episode while still maintaining large batch size, and 2) additional "demo" actors that pulled the current policy parameters and used them to calculate relatively fresh lstm states on the demonstrations.

---

### Official Review · AnonReviewer3 · 2019-10-23
**Official Blind Review #3**

**Rating:** 8

**Review:**

The paper addresses the problem of exploiting human demonstrations in hard exploration (RL) problems. A new set of challenge tasks is introduced that destroys the performance of very strong baseline systems while highlighting the strength of the new system.

The approach (rarely but consistently training on separately prioritized human experience replays) is well motivated by the shortcomings of past agents (either in overfitting the demonstrated solution or only working in environments with not-too-hard exploration challenges). Where work by others have overspecialized on specific challenge environments (e.g. Montezuma's Revenge with weak stochasticity and observability challenges), this work intentionally dives into difficult territory.

This reviewer moves to accept this top-quality RL paper. The new agent, R2D3, is the primary contribution in combining and outperforming previous SOTA agents. These eight new environments are minor contributions with limited potential for impact on the field, but still make an independently positive contribution.

Questions for authors:
- What would be the present-day approximate retail cost for reproducing the experiments in this paper?
- At the action-rate experienced by the human demonstrators (30fps?), how much wall-clock time represented by 40B actor steps? (40 years?) Is this "making efficient use"?
- Does having highly variable initial conditions really force generalization over environmental configurations or is this wishful thinking / mysticism? To make a direct claim about this, the authors should consider an experimental design where certain classes of initial conditions (e.g. starting on the left side of the map) are withheld during training and evaluated only during testing.
- The finding of small demo ratios as being stronger is exciting, but this result seems to be tied to the specific quantity and quality of demonstrations gathered. Could a more general picture of the role of demonstrations be had by ablating the diversity of representations? The 100 demos in the full case might be degraded to 50, 25, 10, etc while holding the demo ratio fixed. This might effectively vary the weight that demonstrations take in the optimization independently of how often distinct demos are actually seen.
- Can these hard-eight scenarios be parametrically scaled up and down in terms of their exploration effort (possibly by just changing the action granularity / movement speed)? With performance on the new benchmark almost saturated in the first paper based on it, there isn't much room to grow here. In the same way that Montezuma's Revenge was found by scanning the culturally-impactful library of Atari games, perhaps more appropriate and lasting challenges can be found by looking one or more generations forward in the history of commercial console games. Can we play Star Fox? What about SimCity?



**Experience Assessment:**

I have published one or two papers in this area.

**Review Assessment: Checking Correctness Of Derivations And Theory:**

N/A

**Review Assessment: Checking Correctness Of Experiments:**

I assessed the sensibility of the experiments.

**Review Assessment: Thoroughness In Paper Reading:**

I read the paper at least twice and used my best judgement in assessing the paper.

---

> ### Author Response · Authors · 2019-11-13
> **Reply to Official Review #3**
>
> > What would be the present-day approximate retail cost for reproducing the experiments in this paper?
>
> We used 256 actors and a single GPU learner. We trained R2D3 approximately for a week on each Hard-Eight task. Based on the numbers provided in Figure 8 of [a] ($0.0475 per cpu per hour, 1.46 per GPU per hour), training R2D3 on a single Hard-Eight task would cost 2288.16 USD.
>
> [a] Seed RL: Scalable and Efficient Deep-RL with Accelerated Central Inference
>
> > At the action-rate experienced by the human demonstrators (30fps?), how much wall-clock time represented by actor steps? (40 years?) Is this "making efficient use"?
>
> Yes, it would take approximately 64 years. In our problem setting, one can consider efficiency with respect to demonstrations and/or environment interactions. We claim that our method can make efficient use of demonstrations but at the cost of a large number of interactions with the environment. However, we still need significantly fewer interactions with the environment than pure RL approaches which have seen no reward in the same 64 year period.
>
> We are hopeful advances in off-policy RL and model based RL will improve the interaction efficiency of RL from demonstrations in the future.
>
> > Does having highly variable initial conditions really force generalization over environmental configurations or is this wishful thinking / mysticism? To make a direct claim about this, the authors should consider an experimental design where certain classes of initial conditions (e.g. starting on the left side of the map) are withheld during training and evaluated only during testing.
>
> Sorry for any confusion. There are two types of generalization we can discuss in this setting: Type 1) generalizing from a small number of demonstrations to all initial conditions in the training task and Type 2) generalizing from initial conditions in the training tasks to the initial conditions in a hold out tasks. Type 1, which is not commonly considered, is the type of generalization we are focused on in this work. We agree that Type 2 is quite interesting but we haven't tested it thoroughly in this work.
>
> > The finding of small demo ratios as being stronger is exciting, but this result seems to be tied to the specific quantity and quality of demonstrations gathered. Could a more general picture of the role of demonstrations be had by ablating the diversity of representations? The 100 demos in the full case might be degraded to 50, 25, 10, etc while holding the demo ratio fixed. This might effectively vary the weight that demonstrations take in the optimization independently of how often distinct demos are actually seen.
>
> This is a good suggestion. We considered this experiment but it is fairly compute intensive to run this. We ran some preliminary experiments on one of the easier tasks (drawbridge), where we varied the number of demonstrations and R2D3 managed to solve drawbridge even with 20 demos. We did not vary the demo ratio (fixed to 1/256), or try the other tasks.
>
> If you think these experiments would be valuable to include, we can include them in the final version of the paper.
>
> > Can these hard-eight scenarios be parametrically scaled up and down in terms of their exploration effort (possibly by just changing the action granularity / movement speed)? With performance on the new benchmark almost saturated in the first paper based on it, there isn't much room to grow here. In the same way that Montezuma's Revenge was found by scanning the culturally-impactful library of Atari games, perhaps more appropriate and lasting challenges can be found by looking one or more generations forward in the history of commercial console games. Can we play Star Fox? What about SimCity?
>
> Regarding scaling exploration difficulty: Yes the levels could be modified in simple ways to make them more difficult. The action repeats or speed as you suggested, or by modifying the levels for example by making the rooms larger. Thanks for the suggestion, we may do this when we release the environments.
>
> Regarding saturating performance: R2D3 achieves the max possible reward for 5 tasks out of 8. However, there are still three tasks that are not completely solved yet. In addition, as was previously noted, these tasks are quite compute / exploration intensive to solve. We believe that the Hard-Eight tasks can be an interesting domain to improve the sample efficiency of RL algorithms. It is also an interesting domain for improved exploration methods.
>
> Regarding lasting challenges: It is quite interesting to use commercial games as a benchmark to test agents. We are aware of the efforts on Starcraft 2, DOTA, and others. However, those games can be even more compute intensive to run. The Hard-Eight tasks present a middle-ground between the current RL benchmark environments and commercial games in terms of compute required to solve with the existing algorithms and difficulty.

---

### Official Review · AnonReviewer4 · 2019-11-01
**Official Blind Review #4**

**Rating:** 6

**Review:**

In this work, R2D3 (Recurrent Replay Distributed DQN from Demonstration), which combines R2D2 [1] with imitation learning (IL), is proposed. Similar to the existing works on “reinforcement learning (RL) with demonstration” such as DQfD, DDPGfD, policy optimization with demonstration (POfD) [2], hard exploration conditions (sparse reward, partial observability, high variance in initial states) are assumed, which is difficult to achieve good performance with RL without demonstration in general. Eight tasks in such conditions were devised and used to test the performance of R2D3.

I like the fact that the authors of this work have chosen quite challenging scenarios, but I think the novelty of this submission is a bit weak to be accepted to the conference. I believe “RL with demonstration” becomes meaningful when it beats both RL and IL in some reasonable setting. For example, POfD [2] assumes sparse-reward tasks with *imperfect* demonstrations, which is difficult to achieve good performance by using RL or IL. From such a perspective, I have the following concerns:

- Imitation learning baselines: There has been recent advancement in imitation learning. In the submission, it was mentioned that “GAIL has never been successfully applied to complex partially observable environments that require memory”, but there’s [3] that successfully uses GAIL in such a setting. Also, off-policy imitation learning such as DAC [4] is shown to be highly sample-efficient compared to GAIL in MuJoCo domain. However, the submission only considers behavioral cloning (BC) (which shows poor performance at unseen states due to the covariate shift problem) as a baseline among imitation learning method

- Reinforcement learning baselines: The submission adopted R2D2 as an RL baseline, and it seems to me that the R2D2 agent starts from random initialization. For a fair comparison, however, I believe R2D2 with BC (or Batch RL) initialization should be considered.

In addition to the above concerns, it seems to me that most of the features in R2D3 simply combines those in either DQfD or R2D2, and I couldn’t find out its own algorithmic novelty except “demo ratio” parameter.

I’ll increase my score if I made wrong comments or misunderstood the contribution.

References
[1] Kapturowski, Ostrovski, Quan, Munos. and Dabney, “Recurrent experience replay in distributed reinforcement learning,” ICLR 2019.
[2] Kang, Jie, Feng, “Policy optimization with demonstrations,” ICML 2018
[3] Gangwani, Lehman, Liu, Peng, “Learning Belief Representations for Imitation Learning in POMDPs,” UAI 2019
[4] Kostrikov, Agrawal, Dwibedi, Levine, Jonathan, Tompson, “Discriminator-Actor-Critic: Addressing Sample Inefficiency and Reward Bias in Adversarial Imitation Learning,” ICLR 2019

**Experience Assessment:**

I have published one or two papers in this area.

**Review Assessment: Checking Correctness Of Derivations And Theory:**

N/A

**Review Assessment: Checking Correctness Of Experiments:**

I assessed the sensibility of the experiments.

**Review Assessment: Thoroughness In Paper Reading:**

I read the paper at least twice and used my best judgement in assessing the paper.

---

> ### Author Response · Authors · 2019-11-13
> **Reply to Official Review #4**
>
> > (Novelty related concerns) … I like the fact that the authors of this work have chosen quite challenging scenarios, but I think the novelty of this submission is a bit weak to be accepted to the conference.
>
> We understand that this work can be seen as incremental from the algorithmic point of view. In that sense, we showed that it is possible to achieve significant improvements on hard tasks with a novel combination of well-known techniques. We believe that this fits well with the acceptance criteria for ICLR. The reviewer guidelines suggests that papers that present SOTA results on well-studied problems should be given consideration, if they address problems that are of interest to the community.
>
> We believe that our work is interesting to the community, because it shows that these challenging tasks can be solved with only a small number of demonstrations.
>
> > (Concern on the imperfect demos) ... For example, POfD [2] assumes sparse-reward tasks with *imperfect* demonstrations, which is difficult to achieve good performance by using RL or IL.
>
> Agreed, RL with demos is very interesting in the imperfect demo setting. Our work also falls into this setting (see the average reward of the demonstrations in Table 1). We clearly demonstrate that RL from demonstrations has beaten both RL and IL in this setting. And thank you for pointing out POfD, We will add it to our related work.
>
> >  (GAIL baseline) … In the submission, it was mentioned that “GAIL has never been successfully applied to complex partially observable environments that require memory”, but there’s [3] that successfully uses GAIL in such a setting.
>
> We will fix that statement. We would like to point out that standard GAIL does not work in the following settings: 1) POMDPs [3], 2) from pixels [a, b], 3) off policy [4] and 4) with variable initial conditions [c]. Let us note that [3] only addresses partially observable environments for GAILs. Our setting combines all of these. Each GAIL extension cited above is nontrivial and combining them may present challenges which are beyond the scope of this work.
>
> [a] InfoGAIL: Interpretable Imitation Learning from Visual Demonstrations. 2017.
> [b] Visual Imitation with a Minimal Adversary. 2018.
> [c] Task-Relevant Adversarial Imitation Learning. 2019.
>
> > (Batch-RL or BC initialized baseline) ... For a fair comparison, however, I believe R2D2 with BC (or Batch RL) initialization should be considered.
>
> Thanks for pointing this out, we considered initialization with BC baseline. However, BC was performing very poorly on the Hard-Eight tasks, due to the small number of demos. As a result, we believed the representation learned by the BC may not be useful for the R2D2. We will add a batch-RL initialized R2D2 baseline to the camera-ready version of the paper.

---

### Author Response · Authors · 2019-11-13
**To All Reviewers**

We would like to thank all the reviewers for their thoughtful comments and feedback. There were multiple questions about releasing the environments. We are planning to release the Hard-Eight tasks before the ICLR conference. We also plan to open source R2D3 in time for the ICLR conference.

---

### Decision · Program_Chairs · 2019-12-19

**Decision:**

Accept (Poster)

**Comment:**

This paper tackles hard-exploration RL problems using learning from demonstrations. The idea is to combine the existing R2D2 algorithms with imitation learning from human demonstrations. Experiments are conducted on a new set of challenging tasks, highlighting limitations of strong current baseline while highlighting the strength of the proposed approach.

The contribution is two-folds: the proposed algorithm which clear outperforms previous SOTA agents and the set of benchmarks. All reviewers being positive about this paper, I therefore recommend acceptance.